# Information and Communications Technology (ICT) Usage during COVID-19: Motivating Factors and Implications

**DOI:** 10.3390/ijerph18073571

**Published:** 2021-03-30

**Authors:** Yi-Ching Lee, Lindsey A. Malcein, Sojung Claire Kim

**Affiliations:** 1Department of Psychology, George Mason University, 4400 University Drive, Fairfax, VA 22030, USA; lmalcein@masonlive.gmu.edu; 2Department of Communication, George Mason University, 4400 University Drive, Fairfax, VA 22030, USA; skim205@gmu.edu

**Keywords:** Information and Communications Technology (ICT), COVID-19, social connectedness, future anxiety, social media, technology and society

## Abstract

This study was designed to investigate the roles information and communications technology (ICT) played during the current COVID-19 pandemic. Specifically, we focused on the relationships between ICT use and perceived importance of social connectedness and future anxiety, while considering relevant personality and psychosocial factors. A U.S. sample of 394 adults answered questions about ICT use, pandemic-related reactions and actions, demographics, and psychosocial factors via an online survey. Using logistic regression, findings indicated that personality (extraversion and conscientiousness) and psychosocial (need to belong and perceived attachment to phone) factors, types of ICT as news source, and gender were associated with perceived importance of social connectedness. Neuroticism, time spent on ICT for social purposes, and perceived threat of COVID-19 were associated with future anxiety. In addition, using Mann–Whitney U test, people who rated higher on importance of social connectedness had higher ICT use, both in terms of types and time spent on ICT. Overall, results are consistent with the idea that technology is a coping tool during the pandemic and balanced use can lead to feelings of social connectedness and less future anxiety. Therefore, it is important for authorities to align their messaging and outreach with people’s psychosocial, personality, and health considerations through ICT channels while empowering ICT users to be responsible for their interactions with the technology.

## 1. Introduction

In early January of 2020 the World Health Organization (WHO) reported that an outbreak of pneumonia-like cases in Wuhan, China, had been determined to be caused by a novel coronavirus. With evidence of human-to-human transmission, the virus started to spread and many nations across the globe started to report their own cases. On January 30th the WHO declared the coronavirus outbreak a Public Health Emergency of International Concern, and on February 3rd, the United States (U.S.) declared a Public Health Emergency due to the outbreak. The disease caused by the novel coronavirus was officially named COVID-19 on February 11th. On March 11th WHO declared COVID-19 a Pandemic [1], and two days later the U.S. declared COVID-19 a National Emergency. As a way to slow the spread of transmission, most countries around the world implemented social distancing, quarantine, and lockdown guidelines. Due to these health and safety measures, global citizens faced unprecedented changes to their daily routines, including stay-at-home orders, travel bans, and closures of educational institutions and entertainment-related locales.

One change affecting many people was in their use of Information and Communications Technology (ICT). ICTs are broadly defined as products that can digitally store, retrieve, manipulate, transmit, or receive information, such as personal computers, televisions, telephones, email systems, robotic and smart devices, and other internet-enabled systems, including traditional media and social media [2,3,4]. For example, with regard to television use, during the week of March 16th of 2020, viewership of the four largest broadcasts networks in the U.S. increased nearly 19% compared to the same week during 2019. In terms of news consumption during the week of March 16th, viewership of cable news networks increased 73% from 2019 to 2020 and increased 40% compared to the week of February 17th of 2020. During the week of March 16th, number of digital visits to news websites in the U.S. increased by 68% and the number of digital visits to government sources (e.g., WHO) increased by 299% compared to the week of February 17th [5]. In regard to cell phone usage, data collected in May 2020 from the U.S. suggested that people have become more dependent on their phone service: 37% increase in texting, 36% increase in social media, 23% increase in the use of shopping apps, and 32% increase in video calling [6]. Similarly, Zoom, Google Classroom, and Microsoft Teams have documented increased use of video calling during the first half of March [7]. Mobile, contactless payments and online food ordering have also seen increased use in tandem with social distancing measures [8,9].

Empirical studies have also reported an increasing trend of ICT use and higher risk of excessive internet use during COVID-19 quarantine or lockdown [10]. The increased use of ICT can ultimately be associated with various physical, psychosocial, and mental health outcomes [11,12]. This increased use may be driven by disrupted daily routine, need for telework and online schooling, anxiety due to uncertainty about the future, and need for entertainment, news, and social connectedness. Individual differences such as personality traits may also affect feelings about and responses to COVID-19 safety measures and social connectedness. This paper aimed to investigate the relationships among ICT use, social connectedness, and feelings about the pandemic. Relevant literature related to the connection between ICT use and COVID-19 as well as the associated feelings about social connectedness and the future will be reviewed in the next section.

## 2. Literature Review

### 2.1. ICT Use during the Pandemic

Engagement with ICT during the COVID-19 pandemic has received mixed reporting. On the one hand, ICT facilitates the dissemination of information and facts about the disease while allowing people to access and search for related updates [13]. Given the evolving nature of the pandemic, the practical challenge is how to best transfer and deliver the latest information efficiently. Traditional methods of dissemination and communication, such as static websites and even email are considered slower than the use of news media and social media [14]. For example, the use of educational materials and infographics via social media has been viewed as a speedier way of information dissemination compared to traditional methods [13,15]. Additionally, large scale working from home and online schooling has become possible due to the use of ICT and other internet-enabled technologies [16,17,18]. Similarly, telehealth services can provide feasible on-going or new treatment options via online means during the pandemic [19,20]. Staying socially connected with families and friends and having access to virtual physical exercise materials and entertainment during stay-at-home orders are realized through ICT [20,21], as these strategies are recommended for mental health by the WHO [22].

However, the use of ICT can also be problematic. Among children and university students, excessive screen time and limited outdoor activities during the pandemic have potential worrisome outcomes in relation to myopia [23], sedentary behaviors [24], disrupted sleep routines [25], and reduced physical activity [26], just to name a few. Adults also report worse depression, loneliness, and stress being associated with increased screen time and reduced physical activity [27]. These lifestyle changes during the pandemic have been linked to poor mental health [27], thus confirming the established association between excessive screen time and negative mental health outcomes from pre-pandemic times [28,29]. Another downside of ICT use is related to the lack of in-person social interactions. Even for people who do not live alone, they may still feel lonely if their contact with others, such as through the means of ICT during quarantines, does not provide a sufficient sense of social connectedness [30]. In addition, with the amount of information about COVID-19 that is available through various ICT channels, some people have expressed feelings of information overload and fatigue [31,32], while, at the same time, having feelings of anxiety and uncertainty about the future and especially about how the pandemic will end [33].

### 2.2. Social Connectedness during the Pandemic

One consequence of COVID-19 and the associated social-distancing measures has been accompanied by psychosocial implications, including increased risk of social isolation and loneliness [30]. Social isolation—the objective lack of interactions with others or the wider community [34]—even as short as 10 days, can have negative long-term effects three years later [35]. Loneliness—the subjective feeling of the lack of social networks or companions [34]—can be triggered by social isolation, or vice versa [36]. Prolonged social isolation and feelings of loneliness, characterized by reduced social connections and contact, have been linked to reduced psychological and physiological functioning and increased morbidity and mortality [37,38]. Therefore, public health agencies and clinicians emphasize the importance of maintaining social contact during this pandemic for the purpose of improving feelings of social connectedness and decreasing loneliness [30].

Psychoactive substance use and other reinforcing behaviors such as video gaming, TV watching, using social media, gambling, and surfing the internet are often used to reduce anxiety and depression [10], but increased ICT consumption can also lead to negative health outcomes, as have been observed during COVID-19 [27,39]. For example, one study investigated the impact of COVID-19 on online gambling during the week of April 21st, 2020 and found that individuals with higher levels of anxiety and depression were more likely to have gambled than individuals with no symptoms [40]. Among a sample of adolescents and young adults across several countries, COVID-related worries, compulsive internet use, social media use, and gaming addiction predicted scores of escapism, depression, and loneliness [41]. A study conducted during June 2020 found that college students had excessive use of social networking sites and lack of personal control to disengage themselves from those sites [42]; this tendency was also associated with the use of alcohol, tobacco, cannabis, and sedative without a doctor’s prescription. Another study conducted in March and April 2020 in the U.S. suggested that exposure to COVID-19 information, via Twitter, Instagram, and Facebook, and increased alcohol use in March contributed to more frequent alcohol consumption in April, especially those working or studying from home [43]. These empirical studies suggest that even though ICT provides a means of social connection with one’s social networks and the wider community, moderate and responsible use is important in maintaining a healthy approach to it.

Personality traits have also been linked to COVID-19 stress, coping, and concerns. Extraverts—compared to introverts—tend to have larger social network sizes [44] and these network connections can serve as a buffer during the pandemic [45]. Arguably, extraverts may suffer more due to COVID-19 travel bans and restrictions on social gatherings. An online study conducted in late March and early April 2020 across 47 countries investigated the associations among the level of stringency of safety measures, extraversion, and depression [46]: Results showed that, after controlling for country-level factors, introverts were doing better in terms of depressive symptoms when facing stringent social-distancing measures, but the stringent measures only had limited, non-significant effects on extraverts’ depressive symptoms. Other aspects of personality traits were also examined: using an adult sample collected in late March 2020 from the U.S., individuals higher on neuroticism and lower on conscientiousness had more pandemic-related concerns, especially health-related concerns; however, individuals higher on neuroticism and extraversion had more relationship-related concerns [47]. Higher conscientiousness was associated with more precautious behaviors (such as hand washing) to avoid contracting COVID-19 but fewer preparatory behaviors (such as buying face masks). However, older individuals who were higher on conscientiousness had more preparatory behaviors. In terms of estimates of the pandemic duration, individuals higher on neuroticism had a more negative feeling about the pandemic (i.e., longer duration), but individuals higher on extraversion and conscientiousness had more optimistic estimates [47]. These findings shed some light on the associations between personality traits and behaviors related to COVID-19; however, other aspects of behavioral and psychosocial responses, such as social connectedness in the context of ICT use and its relation to personality, have not been widely explored.

ICT plays an important role in helping people adapt to restrictions on in-person gatherings. Many organizations in the public, private, and philanthropic sectors have developed messaging and outreach programs to specifically promote social connections and reduce loneliness via emails, websites, or smart phone applications. ICT serves as communication channels and social interaction media between sources of information and receivers. Although online interactions—both on the giving and the receiving ends—can foster a sense of connection, there is conflicting evidence about the role ICT plays in enhancing the feeling of social connectedness during the COVID-19 pandemic. This paper was designed to further investigate this open question.

### 2.3. Feelings About the Future during the Pandemic

At the time of writing, the pandemic is still ongoing and affecting people’s lives holistically. While the prospect of a vaccine provides a sense of relief for some, there is still a lot of anxiety and uncertainty about when and how the pandemic is going to end and whether lives will return to the pre-pandemic normal. These feelings are not easy to tease apart, as they are deeply intertwined with the economic, societal, and psychological consequences of the pandemic [48].

These feelings are also related to the perceived threat of the pandemic. An online study conducted in June 2020 during a lockdown measure investigated the relationships among perceived threat of COVID-19, future anxiety, and subjective well-being [49]. Results indicated that perceived threat negatively predicted subjective well-being, and this relationship was mediated by anxious feelings about the future. Another online study conducted in early May 2020 investigated the relationships among personality traits, perceived stress during the pandemic, perceived threat of contracting COVID-19, and perceived efficacy to prevent COVID-19 [50]. Results showed that higher neuroticism was associated with higher levels of pandemic-related stress, and this relationship was mediated by perceived threat and efficacy. Similarly, higher extraversion was associated with higher pandemic-related stress.

There have also been studies that examined the effect of media exposure on COVID-related fear and worries. An online study conducted in mid-March 2020 investigated the relationships among fear of COVID-19, intolerance of uncertainty, worry, anxiety, personal relevance of the threat, and media exposure (sources of COVID-19 information); results indicated that media exposure through regular and social media, tendency to worry about health, and risk for loved ones predicted increased fear of COVID-19 [51]. Another study conducted in early April 2020 had participants recall their media sources (i.e., government, commercial, foreign, and social media) from late January to February during the pandemic in China while responding to questions about media traumatization and anxiety due to the pandemic. Most of the survey respondents spent 1-3 h per day watching or hearing COVID-19 information and repeated media exposure led to higher levels of anxiety as well as media traumatization [52]. These findings suggest that the subjective feelings about the pandemic, such as anxiety, may be influenced by a number of psychosocial factors, personality traits, and media exposure. This paper was designed to further investigate the influence of media types for news information or social purposes on subjective feelings about the pandemic.

### 2.4. Study Objectives

Based on the past literature, this paper focused on two consequences of ICT use during the pandemic: social connectedness and feelings about the future. Specifically, we explored how ICT use, behavioral and psychosocial responses during the pandemic, personality traits, and demographic factors influenced perceived importance of social connectedness and perceived future anxiety.

## 3. Materials and Methods

An online, anonymous survey was used for this work. This study received the Institution Review Board approval from the authors’ university. This survey was posted on Mechanical Turk during the afternoon of April 21st and the morning of April 22nd, 2020; during this time, 42 states and territories in the U.S. had issued mandatory stay-at-home orders [53].

### 3.1. Participants

Individuals who were Amazon Mechanical Turk workers and held the status of a Master [54] (workers who have demonstrated high performance over time and meet the performance requirements put forth by Mechanical Turk) were invited to participate. Other inclusion criteria included being an adult (18 years of age or older) and residing in the U.S. A total of 402 participants completed the survey and received the compensation of USD 7. Eight of them provided at least one invalid answer to the three attention check questions (e.g., answering 1978 when the survey asked for the current year, answering February when the survey asked for the current month) and were removed from the dataset; therefore, the final sample size was 394.

### 3.2. Procedure

Individuals who chose to take part would first read the consent page and must agree to the requirement of completing the entire survey. Once they indicated consent, they read the instructions as well as the definitions of the terminology (e.g., ICT) used in the survey. The instructions also emphasized that there were no right or wrong answers and that participants were asked to answer the survey questions honestly. Participants saw one question at a time and were encouraged to answer all the questions, although they could skip questions if they chose to. At the end of the survey, participants were encouraged to leave comments and feedback about the survey and report any technical issues during the study. On average, participants took 16 min to complete the survey.

### 3.3. Key Measures

The survey was programed in Qualtrics software (see Appendix A for the survey items). About half of the items were previously validated psychosocial scales: The Need to Belong Scale [55], the Fear of Missing Out Scale [56], Perceived Attachment to Phone Scale [57], Habitual Smartphone/Internet Behavior Scale [58,59], the Self Regulation Scale [60], the Boredom Proneness Scale [61,62], and the Abbreviated version of the Big Five Inventory [63]. The rest of the survey items were developed by the authors and are detailed below.

#### 3.3.1. Information and Communications Technology (ICT)

There were 6 questions about the ICT. ICT was defined as “the integration of telecommunications and computers as well as necessary software, hardware, and audiovisual systems that enable users to access, store, transmit, and manipulate information and to communicate in a digital form.” These questions were about (1) overall ICT devices used on a daily basis, (2) time spent using ICTs for obtaining news on a daily basis, and (3) the sources for obtaining news on a daily basis (e.g., news channels, radio, etc.). These questions were presented twice—for participants to indicate their answers from two time periods: before the pandemic and during the pandemic.

#### 3.3.2. Reactions Related to the Pandemic

There were 3 questions about the reactions related to the pandemic. One question asked participants to rate their feeling after reading or hearing the news about the pandemic. A second question asked whether the participants considered the current pandemic situation a threat to their health and safety. Another question asked whether they think the news conveyed the current pandemic situation correctly.

#### 3.3.3. Actions Related to the Pandemic

There were 4 questions about the actions that have been taken as the result of the pandemic. Two questions asked whether participants took actions about the pandemic and what the actions were (e.g., social distancing). Another set of questions asked for the time spent on applications (e.g., social media, email, etc.) for the purpose of staying connected with their social network as well as the importance of staying connected with friends, family, and social networks.

#### 3.3.4. Demographic Questionnaire

There were 8 questions. These items asked for participants’ age, gender, residence, state of residence, race and ethnicity, education, income, and employment status.

#### 3.3.5. Attention Check Questions

Three attention check questions were included in the survey. They were added to help identify inattentive participants and to provide progress status, as they appeared after each quarter (1/4, 1/2, and 3/4) in the survey.

### 3.4. Analytic Strategy

A few additional variables were calculated to answer the proposed research questions: (1) Types of ICT devices used on a daily basis: participants could check all that apply from 11 options (e.g., Computer for non-internet use, Computer for internet use, Cable TV, etc.) and write in additional items. These types were then added to reflect the overall total types of devices used. These steps were used for the before and during pandemic periods. (2) Types of ICT as news sources about what is happening on a daily basis: participants could check all that apply from 8 options (e.g., Social media, TV news channels, Radio, etc.) and write in additional items. These types were then added to reflect the overall total types of sources. These steps were used for the before and during pandemic periods. (3) Average hours spent on using applications and systems for the purpose of staying connected with social networks on a daily basis: participants used a sliding bar to indicate the hours for social media, telecommunication, and email and had the options to write in two additional items and then indicate the hours. These hours were then averaged across the items to reflect the average hours spent daily for virtually staying connected with social networks.

To answer our first research question, a logistic regression was used to model the relationship between the perceived importance of social connectedness (low vs. high) and the psychosocial, ICT use, and demographic variables. To answer our second research question, a logistic regression was used to model the relationship between participants’ feeling about the future (positive vs. negative) and the psychosocial, ICT use, and demographic variables.

A non-parametric Mann–Whitney U test was used to compare the distributions of responses due to the non-normality of our data distribution. Correlations of the variables were checked and there was no evidence of multicollinearity (all the Spearman correlation coefficients were smaller than 0.6). SPSS version 26 was used for the analyses.

## 4. Results

### 4.1. Sample Characteristics

The sample consisted of 219 men and 175 women in the U.S., with ages ranging from 20 to 76 and the average being 40.89 (SD = 11.21) years. The participants came from all of the states, except Alaska, Arkansas, North Dakota, South Dakota, and Vermont. In terms of primary residence, 190 indicated suburban areas, 125 indicated urban areas, 77 indicated rural areas, and 2 chose other. The majority of the participants identified their race and ethnicity as White (*n* = 307) (61 as Asian, 20 as Black, 12 as Hispanic/Latino/Spanish origin, 11 as American Indian/Alaska Native, 1 as Native Hawaiian/Other Pacific Islander, and 2 as Other). As for education level, most of participants reported having a college degree (*n =* 219), followed by having some college (*n =* 70), having a graduate degree (*n =* 56), having a high school diploma (*n =* 47), and having some high school education (*n =* 2). The annual household income item included five options: most of participants selected the 45 K–70 K (*n =* 115) and 25 K–45 K (*n =* 105) options, followed by the 70 K–110 K option (*n =* 69), <25 K option (*n =* 62), and >110K option (*n =* 43). Most of them currently had a full-time job (*n =* 258) (22 worked part-time, 82 were self-employed, 3 were a student, 27 were unemployed). The statistics of the psychosocial scales are listed in Table 1.

Almost all of the participants (*n =* 391) indicated that they actively took actions about the current pandemic situation (e.g., social distancing, working from home, etc.). The majority of them thought that the news correctly conveyed the current pandemic situation (*n =* 308) and that the current pandemic situation was a threat to their health and safety (*n =* 315). Slightly less than half of the participants rated their feeling about the pandemic upon reading the news as “Positive—it’s going to be ok” (*n =* 181), while the rest felt “Negative—it’s not going to be ok.”

### 4.2. Importance of Social Connectedness

In answering our first research question, one survey item asked about the importance of staying connected with friends, family members, and social networks. Participants rated from not at all important to extremely important. Given the uneven distribution of the rated responses (see Figure 1), this variable was dichotomized to reflect two levels of importance: low (combined from “not at all important,” “slightly important,” and “moderately important”) and high (combined from “very important” and “extremely important”) importance, having *n =* 192 and 202, respectively, in each level. Using this dichotomized importance as the grouping variable, Mann–Whitney U test suggested that types of ICT devices used and types of ICT as news source were higher in the high-importance group for both before and during pandemic periods. Hours spent on ICTs for obtaining news before the pandemic were not different between the low-and high-importance groups; however, hours were higher in the high-importance group during the pandemic (see Table 2).

Subsequently, a logistic regression was used to model the relationship between the perceived importance of social connectedness (low vs. high) and the psychosocial, ICT use, and demographic variables. These variables were entered in three blocks, with block 1 consisting of psychosocial variables, block 2 consisting of ICT use variables, and block 3 consisting of demographic variables. Insignificant variables were removed with each iteration. The final model had a Nagelkerke R of 0.36 and a Hosmer and Lemeshow Test of *χ*^2^ (8, *N =* 394) = 11.55, *p* = 0.17, indicating a good fit to the data. The classification accuracy was 68.80% for predicting low-importance and 76.70% for predicting high-importance, with the overall accuracy being 72.80%. Table 3 shows the logistic regression coefficient, Wald test, and odds ratio for each of the predictors for the importance of the social connectedness variable. Employing a 0.05 criterion of statistical significance, ratings of extraversion, conscientiousness, need to belong, and perceived attachment to phone, types of ICT as news source, and gender had significant partial effects. For each one-point increase on the five-point extraversion and conscientiousness scales there were odds of higher rating of importance of social connectedness by a multiplicative factor of 1.34, and 1.47, respectively. Similarly, for each one-point increase on the five-point need to belong and perceived attachment to phones scales there were odds of higher rating by a multiplicative factor of 1.11 and 1.33, respectively. For each one additional type of ICT that was used for obtaining news there were odds of higher rating by a factor of 1.31. As for the gender variable (women coded as one), women were 2.17 times more likely than men to report higher rating on importance of social connectedness.

### 4.3. Feelings about the Future

In answering our second research question, one survey item asked participants to rate their feelings (positive vs. negative) upon reading about the current pandemic situation. Using this as the grouping variable, the Mann–Whitney U test suggested that types of ICT devices used, types of ICT as news source, and hours spent on ICTs for obtaining news were about the same in both feeling groups before and during the pandemic periods.

The same modeling approach was used to develop a logistic regression model to predict the feelings (negative feeling coded as one) about the pandemic. The final model had a Nagelkerke R of 0.21 and a Hosmer and Lemeshow Test of *χ*^2^ (8, *N* = 392) = 3.21, *p* = 0.92, indicating good fit to the data. The classification accuracy was 82.10% for predicting the negative feeling and 50.60% for predicting the positive feeling, with the overall accuracy being 67.60%. Table 4 shows the logistic regression coefficient, Wald test, and odds ratio for each of the predictors for the feeling variable. Employing a 0.05 criterion of statistical significance, rating of neuroticism, hours spent on using virtual means for the purpose of staying socially connected with people, and viewing the pandemic situation as a threat had significant partial effects. For each one-point increase on the five-point neuroticism scale there were odds of feeling negatively by a multiplicative factor of 1.41. With one hour increase in using social media, emails, etc. to stay socially connected, participants were 1.10 times less likely to rate the feeling as negative. Participants who thought the current pandemic situation was a threat to their health and safety (yes coded as one) were 6.51 times more likely to rate their feeling about the pandemic negatively.

## 5. Discussion

This study was conducted in the early phase of the pandemic in the U.S. and was designed to examine the role ICT played—in terms of daily use for receiving news and staying in contact with social networks—in people’s feelings about social connectedness and future anxiety while taking into account relevant personality and psychosocial factors. The results showed that the use of ICT was associated with a number of personality, health, and social factors.

Participants who rated higher on importance of social connectedness had higher ICT use, both in terms of types of ICT and time spent on ICT. This is consistent with the concept of using technology as a coping tool [30]. Social-distancing measures and stay-at-home orders took away many forms of communications and social interactions, but thanks to technology, some of them can be supplemented by ICT. Our findings also suggest that we need to have a balanced perspective on monitoring ICT use while allowing users to take advantage of the technology [10,20]. Using social media has been regarded as a negative practice as there is evidence of addiction and excessive usage [41,42]. However, when it is used for staying socially engaged and connected during COVID-19, our finding showed that people had less negative feelings about the future. Therefore, meaningful and responsible use of ICT during this pandemic, in particular social media and email, is likely to help people handle the anxiety and stress in the long run.

Prior empirical work suggests that feelings of social connectedness during COVID-19 stay-at-home orders is associated with reduced stress [30,64], and the WHO and other authorities have issued recommendations for enhancing positive feelings about being socially distanced from others [32]. Consistent with results from prior work, our findings showed that individuals higher on extraversion, belongingness, and attachment to one’s phone rated staying socially connected to be more important [45,57,65]. Even though prior work did not study the relationship between conscientiousness and social connection in the context of ICT, individuals higher on conscientiousness tend to be more cautious about health-related behaviors (e.g., exercise more) [66] and take more precautions to avoid contracting COVID-19 [47]. We argue that this approach to health and safety translates to feelings about social networks that individuals higher on conscientiousness may value the importance of and take more active actions about staying connected with others compared to those lower on conscientiousness. In addition, individuals higher on neuroticism had more future anxiety, and this finding is consistent with prior research that neuroticism was associated with more concerns and longer duration estimates of the pandemic [47]. Similarly, individuals who thought the pandemic was a threat to their safety had more future anxiety, and this finding parallels prior work that perceived threat undermines mental well-being [49].

There have been mixed results about the association between gender and COVID-19-related feelings, stress, and actions. For example, women had more COVID-related worries than men but had about the same level of perceived stress and perceived chance of contracting COVID-19 with men [64]. When it comes to using social media to share information about COVID-19, women were 1.58 times more likely to do so than men [67]. Our finding suggests that women were 2.17 times more likely than men to report higher rating on the perceived importance of social connectedness, potentially explaining the reason for women’s higher rate of sharing information on social media.

This study has a few limitations. Arguably, the level and form of human–ICT interaction may be different for each type of ICT [3]. For example, traditional media, such as television channels, may be used for information seeking purposes, whereas social media, such as Facebook, may be used for information seeking purposes as well as maintaining social connectedness. This variability may also differ from one user to another. The current paper broadly defined ICT as digital communication technology that allows users to interact and receive information and did not specify the application environment or users’ prior experience with each ICT. We also did not consider multi-user interactions or devices, such as the case of video gaming applications. In the context of pandemic response, prior research has highlighted the benefits of using digital communication technologies for diagnostic efforts, risk communication practices, and coordination processes [4], and future research should examine the level and form of human–ICT interaction in each of the use categories.

Second, given the nature of data collection, some potential threats to external validity of the study are discussed. First of all, the sample might not be representative of the general U.S. public. The sample size was small and participants were recruited through Mechanical Turk, a crowdsourcing platform developed by Amazon. Walters and colleagues [68] have found that although MTurk workers were similar to the representative national sample, MTurk users tended to be younger, more likely to have a college degree, and less likely to report excellent health status. Additionally, the data were collected around the early phase of the COVID-19 pandemic. With new COVID-19 vaccines and related media, political and economic adjustments, the degree to which the study results will stay the same as the pandemic enters a more advanced stage remains an open question. In addition, only self-reported survey instruments were used; there might have been overreporting, underreporting, or social desirability bias, especially for socially sensitive questions [69]. Although indirect questioning was used to the extent possible (such as the phrasing of third-person wording, as opposed to first-person wording), some direct questioning was unavoidable. Future research should compare these wording differences in the context of a global pandemic and pandemic response in survey-based study design.

Despite the limitations of having a small sample size and the use of self-reported instruments, our findings contribute to the literature by highlighting the role ICTs play during the early phase of the pandemic: they are used for information seeking and social contact. These usage patterns are associated with various feelings about the pandemic. Therefore, it is important for authorities, clinicians, researchers, policy makers, and employers to align their messaging and outreach activities with people’s psychosocial, personality, and health considerations through ICT channels while empowering ICT users to be responsible for their interactions with the technology [70].

## 6. Conclusions

This study was designed to examine the roles ICTs played during the early phase of the COVID-19 pandemic. Adults from a U.S. sample completed questions about their reactions, actions, feelings about the pandemic as well as their personality and psychosocial characteristics via an online survey. We argue that ICTs played multiple roles during the pandemic. Notably, ICTs are broadly defined in this study as the focus is on digital communication technology. Our results suggest that users interact with ICTs for the purposes of information seeking and staying socially connected with their families, friends, and social networks. These findings are in line with the idea that technology is used as a coping tool; however, as the society’s reliance on technology increases during the pandemic and potentially post-pandemic, we need to constructively and mindfully leverage technology to improve our health and safety and reduce anxiety and stress. Users are to pay attention to their usage habits, and responsible interactions with ICT are critical in improving feelings of social connectedness while minimizing feelings of future anxiety.

## Figures and Tables

**Figure 1 ijerph-18-03571-f001:**
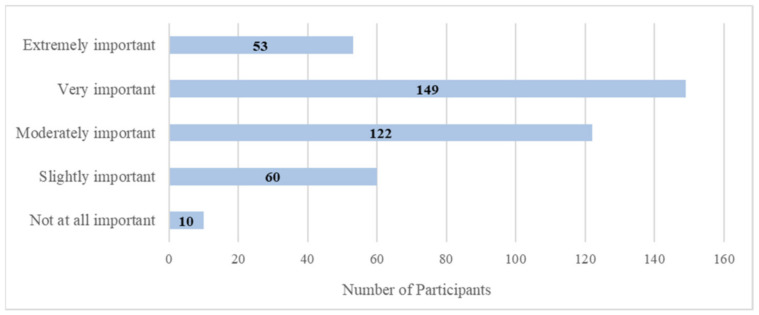
Response distributions for the importance of the social connectedness variable.

**Table 1 ijerph-18-03571-t001:** Statistics of the psychosocial scales.

Scales	Mean	SD	Cronbach’s Alpha
Need to belong	26.77	7.93	0.88
Fear of missing out	20.28	8.18	0.91
Perceived attachment to phones	15.80	5.51	0.87
Habitual smartphone/internet behavior—Smartphone	27.45	7.39	0.95
Habitual smartphone/internet behavior—Internet	31.96	3.52	0.80
Self regulation	31.06	6.03	0.90
Boredom proneness—Lack of internal stimulation	30.95	6.01	0.81
Boredom proneness—Lack of external stimulation	20.12	7.20	0.81
Abbreviated version of the big five—Extraversion	5.49	2.49	0.76
Abbreviated version of the big five—Agreeableness	11.23	2.73	0.67
Abbreviated version of the big five—Conscientiousness	8.17	1.83	0.63
Abbreviated version of the big five—Neuroticism	4.86	2.36	0.79
Abbreviated version of the big five—Openness	7.47	2.06	0.60

**Table 2 ijerph-18-03571-t002:** Comparison of information and communications technology (ICT) use variables between low- and high-importance groups.

	Low-Importance: Mean (SD)	High-Importance: Mean (SD)	U Test	Sig.
Type of ICT use—before pandemic	4.51 (1.67)	4.96 (1.56)	16507	<0.01
Type of ICT use—during pandemic	4.59 (1.68)	5.09 (1.52)	16207	<0.01
Type of ICT as news source—before pandemic	2.43 (1.24)	3.14 (1.53)	14334	<0.001
Type of ICT as news source—during pandemic	3.04 (1.49)	3.99 (1.73)	13352	<0.001
Hours of ICT—before pandemic	2.81 (3.45)	3.12 (3.70)	18414	0.37
Hours of ICT—during pandemic	4.52 (4.10)	5.40 (4.31)	16285	<0.01

**Table 3 ijerph-18-03571-t003:** Significant predictors for the perceived importance of social connectedness variable (*n* = 394).

Variables	B	Wald	Sig.	Odds Ratio
Extraversion	0.31	9.24	<0.01	1.36
Conscientiousness	0.38	7.24	<0.01	1.47
Need to belong scale	0.11	33.33	<0.001	1.11
Perceived attachment to phones scale	0.29	5.98	0.02	1.33
Type of ICT as news source	0.27	12.75	<0.001	1.31
Gender	0.78	10.40	<0.001	2.17

**Table 4 ijerph-18-03571-t004:** Significant predictors for the feeling about the future variable (*n =* 392).

Variables	B	Wald	Sig.	Odds Ratio
Neuroticism	0.35	12.34	<0.001	1.41
Openness	−0.20	3.37	0.07	0.82
Hours spent on virtually staying connected	−0.09	5.32	0.02	0.91
Threat	1.87	35.65	<0.001	6.51

## Data Availability

The data presented in this study are available on request from the corresponding author. The data are not publicly available due to privacy considerations.

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
