# Peer review of "Information and Communications Technology (ICT) Usage during COVID-19: Motivating Factors and Implications"

_ijerph, 2021, doi:10.3390/ijerph18073571_

Round 1

Reviewer 1 Report

The manuscript is well written and organized.

Some suggestions for improvements:

In this article some feelings of IT professionals are discussed.

I suggest adding them in related works: Walter Lucas Monteiro de Mendonça, Pedro Henrique Teixeira Costa, Emille Catarine Rodrigues Cançado, Fernanda Lima, Edna Dias Canedo, Rodrigo Bonifácio, Luis Henrique Vieira Amaral:
From Dusk till Dawn: Reflections on the Impact of COVID-19 on the Development Practices of a R&D Project. SBES 2020: 596-605

There is an errors in Figure 1: (see Error! Reference source not found.).

On line 376 there is also an error: Error! Reference source not found.

What are the threats to validy this research?

What are the limitations?

The conclusion needs improvement, it is very poor.

Reviewer 2 Report

Not only does the article seem appropriate and interesting to society today, but is also appears to me to be rigorous scientific work. 

Its structure is well defined and the concepts well explained. 

Nevertheless, the conclusion section seems too brief. In my opinion, closing information is missing from the article. 

Some more specific insights about the study are:

I think that the size of the sample should be indícate from the abstract, as well as some more information about the results, so that the reader can have an idea of the scope of the work.

I believe, the authors should be more precise when mix terms such as ICTs, with social networks, for example.

In my opinion, this terms are not well differentiated, since in most of the text the authors refer to ICTs, in general, although, in certain paragraphs, they refer to social networks, without even specifying which ones.

Also, dealing with ICTs in a general way is too ambitious, since the behavior of citizens in front of social networks is very different than in front of the television or in front of social networks, or an email, for example.

On the other hand, I consider a determining factor the fact to know more about the respondents, because in the case of the pandemic and, consequently, the circumstances of lockdown or social distance, it is very different if the respondent is a patient, his or her relative, or a person without any type of pathology. More if possible, in the event that he was receiving psychological treatment.

At the same time, I lack more information on the training of the respondents in the use of ICTs, differentiating, I repeat, the information they receive on television, social networks or email, for example, since their characteristics are very different.
